# Evolutionary Analysis of TCGA Data Using Over- and Under- Mutated Genes Identify Key Molecular Pathways and Cellular Functions in Lung Cancer Subtypes

**DOI:** 10.3390/cancers15010018

**Published:** 2022-12-20

**Authors:** Audrey R. Freischel, Jamie K. Teer, Kimberly Luddy, Jessica Cunningham, Yael Artzy-Randrup, Tamir Epstein, Kenneth Y. Tsai, Anders Berglund, John L. Cleveland, Robert J. Gillies, Joel S. Brown, Robert A. Gatenby

**Affiliations:** 1Departments of Integrated Mathematical Oncology, H. Lee Moffitt Cancer Center, Tampa, FL 33612, USA; 2Cancer Biology and Evolution Program, H. Lee Moffitt Cancer Center, Tampa, FL 33612, USA; 3Departments of Tumor Biology, H. Lee Moffitt Cancer Center, Tampa, FL 33612, USA; 4Institute for Biodiversity and Ecosystem Dynamics, University of Amsterdam, Meibergdreef 9, 1105 AZ Amsterdam, The Netherlands; 5Departments of Cancer Physiology, H. Lee Moffitt Cancer Center, Tampa, FL 33612, USA; 6Departments of Pathology, H. Lee Moffitt Cancer Center, Tampa, FL 33612, USA; 7Department of Diagnostic Imaging & Interventional Radiology, H. Lee Moffitt Cancer Center, Tampa, FL 33612, USA

**Keywords:** cancer genetics, evolutionary triage, cancer cell fitness, lung cancer, EGFR, KRAS

## Abstract

**Simple Summary:**

Evolution drives the initiation and progression of cancer. This is apparent in the concept of “driver mutations” that initiate cancer and observed in cells of the lineage. Less appreciated is natural selection’s role in conserving genes that are necessary for optimal cancer cell fitness. We identified highly mutated and highly conserved (under-mutated) genes across subtypes of lung adenocarcinoma distinguished by their driver mutations. The subtypes often shared highly mutated genes suggesting common utility in adapting to similar tissue environments. Conversely, conserved genes were subtype specific indicating tight co-adaptation with the initiating driver mutation. Conserved genes were highly expressed compared to those selected for mutations consistent with our hypothesis that they are critical for optimal fitness. Thus, subtype-specific conserved genes reveal variations critical molecular pathways and cellular functions within each tumor subtype. Computer simulations suggest targeting tumor-specific conserved genes may represent a highly effective treatment strategy. More generally, we present an investigative approach that uses evolutionary selection for hypotheses building and to identify genes in which further investigation should yield maximal clinical benefit.

**Abstract:**

We identify critical conserved and mutated genes through a theoretical model linking a gene’s fitness contribution to its observed mutational frequency in a clinical cohort. “Passenger” gene mutations do not alter fitness and have mutational frequencies determined by gene size and the mutation rate. Driver mutations, which increase fitness (and proliferation), are observed more frequently than expected. Non-synonymous mutations in essential genes reduce fitness and are eliminated by natural selection resulting in lower prevalence than expected. We apply this “evolutionary triage” principle to TCGA data from *EGFR*-mutant, *KRA*S-mutant, and NEK (non-*EGFR/KRA*S) lung adenocarcinomas. We find frequent overlap of evolutionarily selected non-synonymous gene mutations among the subtypes suggesting enrichment for adaptations to common local tissue selection forces. Overlap of conserved genes in the LUAD subtypes is rare suggesting negative evolutionary selection is strongly dependent on initiating mutational events during carcinogenesis. Highly expressed genes are more likely to be conserved and significant changes in expression (>20% increased/decreased) are common in genes with evolutionarily selected mutations but not in conserved genes. *EGFR*-mut cancers have fewer average mutations (89) than *KRAS*-mut (228) and NEK (313). Subtype-specific variation in conserved and mutated genes identify critical molecular components in cell signaling, extracellular matrix remodeling, and membrane transporters. These findings demonstrate subtype-specific patterns of co-adaptations between the defining driver mutation and somatically conserved genes as well as novel insights into epigenetic versus genetic contributions to cancer evolution.

## 1. Introduction

We use differences in patterns of Darwinian selection for critical genes in mutant *EGFR* (*EGFR*-mut), mutant *KRAS* (*KRAS*-mut), and non-*EGFR/KRA*S (NEK) lung cancers to identify variations in their evolutionary arcs and associated co-adaptations that govern their molecular characteristics [1,2,3]. Following an extensive review of molecular oncology literature, Magalhaes [4] found the majority of human genes can be associated with cancer and concluded: “the challenge is determining which are the key drivers”. Similarly, it is increasingly recognized that epigenetic alterations in the expression of normal, non-mutated genes also contribute to cancer evolution [5]. These essential genes cannot be identified on mutational screens, but theory and mathematical models suggest disrupting their function may be an effective treatment strategy [1].

We hypothesize that discerning evolutionary selection for both mutated and conserved genes within different tumor types can be used to demonstrate clinically valuable therapeutic targets. In evolving populations, each mutation in coding genes is subject to natural selection that determines its subsequent prevalence in a population [1,6]. Various mathematical methods have been applied to the underlying evolutionary dynamics of tumor growth [6,7,8,9,10]. Information theory [6,11] and game theory [1], investigation linking molecular changes to carcinogenesis predicted the mutational frequency of a gene within a tumor or cohort of tumors represents an observable manifestation of intra-tumoral natural selection. Thus, when a mutation increases fitness, the affected cell will proliferate at the expense of less fit cells (“positive” or “directional” selection) so that it is observed more frequently than expected by chance. Mutations that do not alter proliferation (“neutral drift”) will have a prevalence determined by chance based on the underlying mutation rate. These represent the well-known dynamics associated with “driver” and “passenger” mutations [12,13]. To these dynamics, we add mutations that decrease fitness (and proliferation), which will disappear from the population or persist at low frequencies (“negative” selection). Thus, a gene observed to have fewer mutations than expected by chance alone is likely under negative selection because its current normal function optimizes fitness and cannot be improved through mutations.

These dynamics permit an inverse problem approach in which the observed frequency of mutations in a gene compared to that expected by chance alone, provides evidence for the type of selection.

Prior investigations across multiple cancer types have found relatively few genes under stabilizing selection [14,15,16]. However, based on computer simulations, we hypothesized that conserved genes primarily emerge from co-adaption with the salient driver mutation(s). Thus, conserved genes will vary depending on both the tissue of origin (i.e., proliferative constraints to malignant growth in local tissue acting as selection forces) and the initiating genetic mutations during carcinogenesis [17]. The tissue environment and the cancer initiating mutations should influence the ecological value of all subsequent molecular changes in the evolving population of cancer cells [2,6].

In our analysis, we assume that the probability of mutation in each base pair is approximately equal across the genome. This is unlikely to be completely true, but a good starting approximation for our analyses. While some studies ascribe variation in the observed frequency of specific genes to differences in their mutational rate [18,19,20], we note that natural selection alone can result in these observed variations. Furthermore, even if some variations in the mutation rate across the genome do exist, natural selection is always the final arbiter of the prevalence of the mutations based on their contribution to fitness. It is, of course, possible that altered mutation rates in neutral or passenger genes (i.e., one that does not alter fitness) could manifest as a difference in prevalence. However, this should result in consistent increased prevalence in all cancer subtypes. In contrast, we find only a small number of genes with an increased mutation rate in all the lung adenocarcinomas (LUAD) subtypes and no overlap among conserved genes.

Here, we investigate the influences of tissue of origin and initiating driver mutations on evolutionary selection for mutated (directional selection) and non-mutated genes (stabilizing selection) during the evolutionary arcs of lung adenocarcinomas. These tumors must adapt to common growth constraints generated by the lung environment but emerge from different initiating molecular events: *EGFR* mutation (*EGFR*-mut), *KRAS* mutation (*KRAS*-mut), and no *EGFR* or *KRAS* mutation (non-*EGFR/KRA*S).

By identifying conserved genes, this investigative approach demonstrates molecular pathways and cellular functions most critical for maintaining malignant growth in each LUAD subtype. Prior computer simulations predict that disrupting highly conserved genes can produce therapeutic effects equivalent to targeted therapy for driver mutations [1] so that uniquely conserved genes in each subtype may represent novel and tumor-specific therapeutic targets.

## 2. Methods

### 2.1. Gene List Acquisition

We divided the TCGA LUAD cohort and classified patients based on known driver mutations in KRAS (G12, G13, Q61, A146), BRAF (V600, N581, G464, G466, G469, G596, D594), and EGFR (L858, S768, L861, G719, T790, indels in exons 18–21). Samples were excluded if they matched criteria for more than one of these genes. Samples that did not meet any of the mutation criteria were classified as NEK (non-EGFR/KRAS). This resulted in three cohorts: EGFR-mut LUAD (*n* = 58), KRAS-mut (*n* = 163) and NEK(*n* = 313), with a total sample size of 534 patients, (downloaded March 2016) [21].

TCGA data was whole exome sequenced with paired tumor/normal analysis to exclude germ line mutations. For our analysis, we limit the genes and associated mutations to those that alter amino acid sequencing (non-synonymous) or those that alter the number of amino acids in the sequence (truncation, stop-gain, frameshifting indel, splicing). Tumor and normal sequence alignment files were downloaded from the Genome Data Commons, and gene-level depth of coverage was calculated by calculating bases covered by sequencing from the above files across each of the RefSeq coding genes (with 25 base pair flanking regions). A base was considered sufficiently covered if the depth of coverage was ≥ 14 in tumor sample and ≥ 8 in normal samples (as has been previously described: https://www.synapse.org/#!Synapse:syn1695394, accessed on 11 March 2016). The fraction of each gene’s protein coding bases (using the longest RefSeq transcript) covered by sufficient sequence data was calculated for each sample using the Negative Storage Model [22].

Coverage files were downloaded as described in the Synapse page. Gene-level depth of coverage measures the fraction of each gene (longest transcript) covered by sequencing data. To address sequencing artifacts that falsely decrease mutation rates, we excluded genes with low average depth of coverage (< 50%) and errors in the RefSeq gene model.

To minimize potential artifacts related to expression, and to focus on genes that are likely functional, genes were analyzed only if their expression is > 2.0 log_2_ in at least one sample (either normal tissue or 1 of the 3 LUAD cohorts). Changes in expression in one or more tumor samples compared to normal tissue were analyzed separately to supplement the genetic data.

### 2.2. Mutational Frequency

Our approach identifies genes that are mutated more or less frequently than expected based on chance alone. This is established by plotting the observed number of mutations in each gene for each cohort against its size (number of base pairs).

Our primary metric for natural selection was based on whether mutations to a gene were less (stabilizing selection) or more (directional selection) frequent than expected by chance. Assuming the probability of mutation was approximately equal for every base pair in an expressed gene, the background mutation rate was determined by regressing the mutational frequency of each gene within each subtype against gene size (Figure 1a). The distance of each gene to the regression line was then determined, and this standardized residual was compared across subtypes (Figure 1b–d). A negative residual value indicates fewer mutations whereas a positive value indicates more mutations than expected.

Within each subtype, evidence for stabilizing or directional selection was defined as those with a distance from the neutral line > 2 standard deviations below or above the mean, respectively (Appendix A). However, this approach was limited in the *EGFR*-mut cohort, which was both smaller in size and had a lower overall mutation rate than the other cohorts. Thus, most genes in the *EGFR*-mut cohort have 0 mutations so that, particularly in small genes, evidence for selection cannot be distinguished from chance alone. Similarly, a single random mutation in a small gene in one cohort member could appear significant.

To include the *EGFR*-mut cohort in our study, we applied additional metrics. First, we examined the mutational frequency in each cohort reasoning that the value of a mutation can also be estimated based on its observed prevalence in the population (e.g., EGFR-mut is observed in all members of its eponymous cohort). For the purposes of our analysis, we arbitrarily defined mutations under directional selection if observed in > 10% of cohort members (Appendix A). The mutational frequency data for each cohort is available in Appendix A so that other criteria can be investigated. Similarly, in identifying potentially targetable conserved genes in the *EGFR*-mut cohort, we reasoned that a conserved gene would be most valuable as a potential target if it was both conserved and highly expressed in all cohort members. Thus, we defined evolutionary conservation as genes with 0 mutations and > 4-fold increased expression in the cohort compared to normal lung tissue (Appendix A). For comparison, we also applied these additional metrics to the NEK and *KRAS*-mut cohorts.

### 2.3. Expression Data

RNAseq data for each gene in each tumor subtype as well as adjacent normal tissue was obtained from the TCGA database. To avoid artifact due to genes that are not expressed or under-expressed, we analyzed only genes with > 2.0 expression in the tumor or normal tissue.

### 2.4. Identifying Conserved Pathways and Functions

When investigating groups of genes, curated lists were entered into DAVID (Database for Annotation and Integrated Discovery), available at the website https://david.ncifcrf.gov (accessed on 17 August 2021). Both Gene Ontology (GO)- DIRECT and -FAT were used to identify significant ontologies, including Biological Process (BP), Cell Compartment (CC), and Molecular Function (MF). We then performed functional annotation clustering. We selected clusters based on their being many genes or based on biological significance. Finally, to identify genes in major pathways, KEGG pathways were searched using DAVID. Venn diagrams were constructed using the Ghent University VIB/UGent Center website: http://bioinformatics.psb.ugent.be/webtools/Venn/ (accessed on 1 September 2021).

In identifying key interactome pathways, we assumed cancer cells would be highly intolerant to such disruptions and we thus considered a gene to be conserved if it had 0 mutations in members of the entire subtype and had normal or increased expression in LUADs.

## 3. Results

### 3.1. Cohort Demographics

Demographic details were missing or incomplete in about 20% of members in each cohort. The available data are shown in Appendix A. Patient age and tumor stage at diagnosis were not significantly different in the 3 cohorts. Consistent with prior studies, patients with EGFR-mut lung cancer were far more likely to be female and non-smokers than in the other cohorts.

### 3.2. Gene Expression and Evolutionary Selection

Prior studies in yeast and the human germ line [23] have suggested an increased mutation rate in highly expressed genes concluding that transcription is associated with DNA damage. Here, we find the opposite—highly expressed genes are also highly conserved (Figure 2). This is evolutionarily sensible because highly expressed genes are likely to be critical for optimal cell fitness and, therefore, more likely to be subjected to evolutionary conservation. Supporting this, we find (Figure 2) highly conserved genes rarely show a significant change in gene expression (compared to normal lung tissue) suggesting they are simply maintaining their usual function while genes under evolutionary selection for mutations frequently demonstrate increased or decreased expression (perhaps related to gain or loss of function mutations).

### 3.3. Highly Mutated Genes under Directional Selection

The role of gene size (number of base pairs) in its observed frequency has been noted in several publications. Genes that are observed to be frequently mutated are often labeled drivers, while others [24] propose this results in “false positives” in larger genes in which the increased mutation is simply the result of the larger number of base pairs subjected to random mutation, Thus, for example, genes such as *CSMD*1 and *MUC16* have been labelled cancer drivers and false positives. Here, we show (Appendix A) that despite their large size, both genes are far (> 2 standard deviations) from the neutral line and thus can be appropriately labeled drivers. In contrast other large genes such as *MLL2* and *LRP2* are frequently mutated in the *KRAS*-mut and NEK LUADs but not more than expected based on size and are thus under neutral selection.

Since our focus is primarily evaluation of conserved genes (stabilizing selection), here we present only those over-mutated genes found in > 10% of tumor subtype samples. As has been previously noted, the number of observed mutations expected by chance alone will depend on the mutation rate and gene size. This is summarized in Appendix A in which we compare the frequency with which a gene is mutated in relation to its distance from the neutral line. Thus, all frequently mutated genes in the EGFR-mut cohort are > 2 standard deviations above the neutral line indicating positive (directional) evolutionary selection. In the other 2 cohorts, most, but not all of the frequently mutated genes, are also under directional evolutionary selection. Other mutations under strong directional selection can be found in Appendix A. By this criterion, *EGFR*-mut cancers, consistent with the cohort’s overall decreased mutation rate, have only 10 gene mutations under directional selection compared to 153 and 205 for *KRAS*-mut and NEK, respectively. Among these highly mutated genes, all are frequently mutated and, using the criterion of > 10% prevalence in the cohort, 7 are common to all cohorts, and 2 are common to *EGFR*-mut and NEK. In addition, 109 are common to *KRAS*-mut and NEK so that only 1, 37, and 87 highly mutated genes are unique to *EGFR*-mut, *KRAS*-mut, and NEK tumors, respectively.

The highly mutated genes common to 2 or 3 cohorts include well-known genes (e.g., *TP53, STK11*) and others not extensively investigated. Although *EGFR*-mut had only 10 gene mutations in > 10% of cohort members, all 10 (*TP53, MUC16, CSMD1, RYR2, FLG, PCLO, AHNAK2, GRIN2A, PKD1L1, LAMB4*) show strong directional selection in the other subtypes (although not all met the > 10% criteria; Appendix A). These are enriched for genes associated with Ca^2+^ signaling including ryanodine (*RYR2*) and glutamate (*GRIN2A*) receptors. The critical role of calcium dynamics in LUADs is evident in other common mutations including *PCLO*, associated with calcium channel CaV1.2 [25]; *AHNAK*, with Ca^2+^ voltage gated channels; and *PK1D1,* which regulates Ca^2+^ dynamics in cilia [26]. In addition, *CSMD1,* a membrane bound complement inhibitor, and *FLG* and *MUC16* (CA125) reflect directional selection on genes associated with extracellular inflammation and matrix (ECM) components (see below).

### 3.4. Evolutionarily Conserved Genes as Evidence for Stabilizing Selection

We identified 22, 160, and 248 conserved genes using the criterion of > 2 standard deviations below the neutral line in the *EGFR*-mut, *KRAS*-mut, and NEK subtypes, respectively (Appendix A). By this metric, no conserved genes were common to all three subtypes and only 14 were found in more than one subtype including *REV3L* in *EGFR*-mut and *KRAS*-mut; *HTT* and *VPS13D* in *EGFR*-mut and NEK; *MDN1, SPATA31A7, NBPF10, LRRC37A2, NPIPB5, NBPF11, SPATA31C2, ZNF729, RGPD6, RIMBP3,* and *CACNA2D2* in *KRAS*-mut and NEK. Interestingly, most of these conserved genes have not been extensively investigated in LUAD, although some have been identified in broad molecular cancer risk and prognosis studies [27,28,29]. Nevertheless, our conservation and expression data suggest these genes have been activated and are useful and perhaps essential for their normal, unaltered function.

Conservation of two members of the NBPF (neuroblastoma break point) and SPATA (spermiogenesis associated ATPase) families show a pattern where genes under stabilizing selection have no known role in normal lung tissue while being associated with other tissues particularly the brain and testes. This pattern is evident in the other common conserved genes: *RIMBP3*, associated with manchette function and spermiogenesis [30], *HTT*, mutated in Huntingtin’s disease [31] (although a reduced incidence of cancer is reported in Huntingtin’s disease [32]), *CACNA2D*, associated with Parkinson’s disease [33], and *VPS13D*, associated with ataxia [34] with a possible role in mitochondrial fission and peroxisome biogenesis [30,35].

We identified 446, 160, and 66 genes with no mutations and > 4-fold increase in expression for *EGFR*-mut, *KRAS*-mut and NEK subtypes, respectively (Appendix A). Only 15 genes meeting these criteria were common to all subtypes. Some have known roles in LUAD including: *SLC2A1* (*GLUT1*) [36], *RHOV* [37], *GJB2* [38], *S100A2* [39], *MB* [40], *BARX2* [41], and *CDKN3* [42]. This supports further investigation of the other conserved genes in this category (*SPINK1, FGF11, CCNB2, XAGE1, PRSS3, DDIT4L, FDCSP*) which are not known to have a role in LUADs.

### 3.5. Evolutionary Selection on Cellular Pathways

Sustained proliferation of cancer cells depends upon persistent delivery of oncogenic signals from driver genes. This requires both gain of function mutations in the driver and evolutionary conservation of the molecular circuitry that transmits the signal to other cellular components. Importantly, different evolutionary dynamics will be observed for cancers initiated from mutations of tumor suppressors. Here, absence of signaling due to loss of function mutations in a tumor suppressor eliminates selection pressure on the downstream molecular wiring. As a result, these genes will exhibit a neutral evolution.

Accordingly, the pattern of mutations in the interactomes of driver and tumor suppressor genes can provide insights into critical signaling pathways in LUAD subtypes.

### 3.6. TP53 Interactome in Lung Cancer

*TP53* is active in multiple different cellular pathways and the most frequently mutated gene in LUAD. It is usually viewed as a tumor suppressor requiring a loss of function mutation. As noted above, following a loss of function mutation, the molecular wires in the interactome should exhibit a neutral pattern of mutations. This is, in fact, observed in some genes within the *TP53*-mut interactome (Appendix A). However, there is consistent conservation (under-mutated) in the interactome genes associated with senescence, apoptosis, invasion, DNA repair, metabolism, and mitosis. This suggests mutant P53 may retain some functions that promote the proliferation and survival of the cancer cells, consistent with some recent experimental observations [43].

### 3.7. EGFR-Mut and KRAS-Mut Signaling

Following a gain of function driver mutation, we expect strong stabilizing selection for the molecular wires that carry the oncogenic signals and downstream effectors that perform the necessary functions. Interestingly, within *KRAS*-mut interactome [44], just 2 genes, *WDR20* and *VT1B*, (Appendix A) have 0 mutations, whereas 11 genes in the *EGFR*-mut interactome, *ENGASE, NDUFA4, AP2A1, AP2A2, AP2B1, AP3M1, ERBB2, CCDC37, DNAJA2, GRB2*, and *HSPA5,* have 0 mutations. In total, conserved members of the interactome and downstream effectors in the *EGFR*-mut interactome include 259 protein-coding genes in its eponymous group compared to 34 in the *KRAS*-mut group.

### 3.8. Evolutionary Selection on Cellular Functions

Genes identified as under natural selection by the above criteria are shown in Appendix A. Here, because of space limitations, we present brief analysis of some broad cellular properties.

### 3.9. Evolutionary Selection on Signaling Pathways

In the *EGFR*-mut tumors, 7 members of the MAPK pathway are under stabilizing selection (Table 1). *HRASLS,* a wild type *KRAS* effector [45] and possibly a Ca^2+^-independent N-acyltransferase [46], is conserved. A notable surprise is conservation of *EGF* with > 4-fold increased expression only in *EGFR*-mut LUAD. We can find no prior literature on this topic but our results suggest *EGF* may contribute to the fitness of these lung cancer cells perhaps though complementary signaling circuits. Cancer cells in both *KRAS*-mut and NEK groups conserved members of the MEK family while WT tumor also conserved *MAP4K5*, which does not have a known role in LUAD. *KRAS*-mut and NEK tumors conserved members of the MEGF family, which have not been extensively investigated. *KRAS*-mut cancers conserved *VEGFC*, an activator of lymphangiogenesis and immunomodulator associated with poor prognosis in LUAD [47].

In all 3 LUAD subtypes, *RHOV* is conserved (using the 0 mutations and > 4-fold increased expression criteria). Other conserved genes common to all three LUADs involve members of the Rho Guanine Nucleotide Exchange family, components of the RAB pathway, Interleukin, Ephrin, fibroblast growth factor, and G-couple protein pathways. NEK tumors conserved elements of interferon signaling. All subtypes exhibited directional selection for mutations in lipid receptors.

Evolutionary divergence between LUADs provides evidence for distinct co-adaptations between the initiating driver mutations and subsequent evolution. For example, *RYR3* was highly mutated in *KRAS*-mut and NEK tumors but conserved in *EGFR*-mut tumors. While *RYR2* exhibits directional selection in all cohorts, *RYR1* was highly mutated in just *KRAS*-mut and NEK tumors. In Ephrin receptors, *EFNA4* was conserved in *EGFR*-mut and *KRAS*-mut tumor while *EFNA3* was conserved in *EGFR*-mut and NEK tumors. *EGFR*-mut and *KRAS*-mut conserved *TMEM184A*, a heparin receptor that may regulate angiogenesis [48].

### 3.10. Evolutionary Selection on DNA Repair, Phenotypic Plasticity, and Epigenetic Modifications

Consistent with the low number of average mutations per sample, we find (Table 2) that the *EGFR*-mut cancers conserved 16 genes related to DNA repair (e.g., *BRCA1* and *BRCA2*). In contrast, we identified only 1 and 3 conserved genes associated with DNA repair in the NEK tumors and *KRAS*-mut groups, respectively, suggesting a classic “mutator phenotype”. We hypothesized the role of epigenetic mechanisms for phenotypic plasticity will vary inversely with the mutator phenotype. Consistent with this, we found that *EGFR*-mut cancers conserved more homeobox genes and genes related to RNA Polymerase II compared to *KRAS*-mut and *NEK* cancers (Table 2). In contrast, NEK tumors showed greater selection for mutations in homeobox and RNA Polymerase II genes. Interestingly, NEK tumors uniquely conserved 4 members of the CDY family, which contain a chromodomain and a histone acetyltransferase domain. One prior study found high expression of *CDYL* correlated with poor survival in NSCLC [49]. We find directional and stabilizing selection for genes involved in z-finger proteins in *KRAS*-mut and NEK, but not *EGFR*-mut, subtypes.

### 3.11. Evolutionary Selection on Microenvironmental Interactions

Extracellular Matrix (ECM)-related genes were variously under both stabilizing and directional selection in lung cancer (Table 3). Significant differences in the subtypes suggests that adaptations for modulating the microenvironment are determined by the driver mutations. For example, 8 collagen (COL) family members are either conserved or highly mutated in *EGFR*-mut cancers. In *KRAS*-mut and NEK cancers no genes of the COL family were conserved, while 9 and 11 genes were highly mutated, respectively. All tumor types exhibited strong directional selection for mutations in members of the cadherin and protocadherin families. NEK tumors conserved a single protocadherin family member (*PDCHGB5*) and *EGFR*-mut tumors conserved *CDH3* and *CDH17*.

Mucins are known to evolve in adenocarcinomas and can form complexes that alter signaling circuits [50]. Accordingly, mutations in *MUC16* (CA125) were selected for in *EGFR*-mut and *KRAS*-mut cancers while only mutations in *MUC17* were selected in NEK. *MUC21* and *MUC13* were conserved only in *EGFR*-mut cancers.

Proteases are necessary for ECM remodeling and *EGFR*-mut cancers conserved 3 members of the MMP family. In contrast, no members of the MMP were conserved in *KRAS*-mut and NEK cancers but several member genes were highly mutated. Directional selection for mutations were observed in multiple members of the ADAMTS family in all 3 LUADs. Only in *KRAS*-mut cancers was a single family member, *ADAMTS8*, conserved. *EGFR*-mut and *KRAS*-mut cancers, but not NEK cancers, had both stabilizing and directional selection among genes of the membrane protease family TMPRSS. Finally, members of the CST and SPINK protease inhibitor families were conserved across subtypes. By way of divergence in co-adaptations, *EGFR*-mut and *KRAS*-mut tumors conserved *CST1* and *CST2* while the NEK tumors conserved *CST6* and *CST7*.

### 3.12. Evolutionary Selection on Membrane Proteins

We hypothesized that conserving and upregulating genes associated with membrane transporters would enhance cancer cell fitness by increasing substrate harvest rates as well as providing useful information on substrate availabilities. Common to all subtypes (Table 4) was stabilizing selection for transporters of glucose (*SLC2A1* [*GLUT1*]) and neutral amino acids (SLC7 family). *EGFR*-mut cancers conserved 2 members of the SLC6 family which transport dopamine. NEK cancers conserved 2 members of the *SLC44* family which transport choline. Of note, *EGFR*-mut cancers conserved transporters for monocarboxylate and riboflavin. *KRAS*-mut cancers conserved a Ca^++^ transporter and anion transporter (*SLC26A8*) thought to be expressed only in sperm. NEK cancers conserved additional transporters for neutral amino acids, folate, carnitine, and phosphate.

Evolutionary changes to cell membranes and mitochondrial [51] transmembrane potential are frequently seen in cancers [52]. Accordingly, mutations in calcium voltage dependent channels are under strong directional selection (Table 4) in all 3 LUADs while NEK and *KRAS*-mut tumors conserved *CANNA2D2*. Interestingly, the multifunctional, Ca^++^-binding *S100A2* gene was one of few genes conserved in all 3 cohorts and *S100P* was conserved in the NEK and *KRAS*-mut cancers.

In contrast, there is extensive selection for mutations in K^+^ voltage dependent channels across the cohorts with *KCNQ5* conserved in *EGFR*-mut, *KCNN4* in *KRAS*-mut, and *KCNS1* and *KCNC3* in NEK tumors.

Interestingly, all 3 cohorts conserved the gap junction (by 0 mutations and > 4-fold increased expression) gene *GJB2. GJB6* was conserved only in *EGFR*-mut and *KRAS*-mut cancers while the latter also conserved *GJB3*. Increased expression of *GJB2*, which primarily functions as an ion channel [53], is associated with a poor prognosis in LUAD [38]. *GJB6*, which encodes connexin 30, is often up-regulated in early stage LUADs [54].

All 3 cohorts demonstrated directional selection for genes in the cancer-testes MAGE and XAGE families. The high prevalence of genetic alterations to cancer-testes genes has been noted, although their precise roles remain undetermined [55].

Finally, we note strong evolutionary selection on ABC efflux pumps, which can confer resistance to treatment, even at the time of presentation (i.e., prior to therapy) suggesting pre-treatment evolutionary dynamics may confer de novo treatment resistance.

## 4. Discussion

Here, we identify natural selection by establishing a mutation rate in each LUAD subtype through a linear regression that establishes an expected number of mutations in each expressed gene based on the gene size and the mutation rate in each cohort. A gene in which the observed number of mutations is significantly higher or lower than this is then likely to be under positive selection or negative selection. Thus, for example, *CSMD1* and *CSMD3* are large genes that have been identified as “drivers” in LUADs but also labeled as “false positives” due to their size. Here, we demonstrate the increased mutational frequency is greater than expected even accounting for their large size indicating evolutionary selection. In contrast, other large genes (e.g., *MLL2*, *LRP2*) are frequently mutated but do not fulfill criteria for evolutionary selection.

We note that our methodology assumes a roughly equal probability for mutations in all base pairs and, therefore, differs from prior studies that propose regional variations in mutation rate throughout the genome [24]. We acknowledge the technical excellence of this approach but also note it does require assumptions (e.g., all cancer cells have replication time identical to that of HeLa cells) and methods (e.g., using expression data from the average of 91 cell lines) that may limit application.

Thus, for example, this approach finds genes with increased expression have increased mutation rate, perhaps due to the biomechanical stress of frequent transcription. We find the opposite—genes that are highly expressed are more commonly conserved (Figure 2). Evolutionarily, this is sensible as highly expressed genes are likely critical for optimal fitness and thus more likely to be subjected to evolutionary conservation. This conclusion is supported by the observation (Figure 2) that conserved genes are less likely to significantly change expression compared to those in which mutations are highly selected. However, we acknowledge that other co-variates [56] could produce intra-genomic variations in mutation rate that may introduce errors in our investigation.

Our data demonstrate that initiating genetic events (*EGFR*-mut, *KRAS*-mut or variable in NEK) influence the subsequent pattern of stabilizing and directional selection on other genes and molecular pathways consistent with driver-dependent coadapted syndromes for each cancer type.

We find substantial but incomplete overlap among the cancer subtypes in the highly mutated genes suggesting directional selection to overcome a relatively fixed number of growth constraints in lung tissue. In contrast, conserved genes are rarely shared among the driver-gene-defined subtypes suggesting fitness contributions unique to each subtype. Primary roles of conserved genes appear to include: 1) Maintaining the molecular wires and downstream effectors for oncogenic signals from driver mutations; 2) Stabilizing selection for molecular pathways and cellular functions which represent convergent and divergent evolutionary dynamics.

We find *EGFR*-mutant lung cancers require few additional mutations to evolve a malignant phenotype. One plausible hypothesis is that any lung cancer must overcome a relatively fixed number of growth constraints common to all lung tissue. Conservation of multiple genes in the relatively large *EGFR*-mut interactome suggests broad propagation of its oncogenic signals reduces the need for additional tumor promoting mutations. In contrast, the *KRAS*-mut interactome is both smaller and has fewer conserved genes suggesting *KRAS*-mut cancers must compensate for limited oncogenic signal propagation through an increased number of additional tumor-promoting mutations. In other words, a lung cancer requires an *EGFR* mutation plus X addition gene mutations *or* a *KRAS* mutation plus Y additional gene mutations where Y > X. WT tumors, lacking any driver genes, require still more mutations (i.e., > Y mutations) to generate a malignant phenotype. We note this hypothesis leads to a clinical prediction: the large size of the conserved *EGFR*-mut interactome is consistent with clinical observations that targeted therapy blocking the *EGFR*-mut oncogenic signal profoundly decreases the fitness of individual tumor cells. Resistance may be achieved rapidly if the *EGFR*-mut signal can be replaced (e.g., by a T790M mutation); but, in the absence of such a mutation, rebuilding the network will require extensive molecular rewiring resulting in a durable response [57]. In contrast, the smaller *KRAS*-mut network suggests response to targeted therapy will be less complete and durable than with *EGFR*-mut.

We hypothesize a requirement for more mutations to address a fixed number of environmental selection forces accounts for observations that *EGFR*-mut LUADs occur in younger patients than the other subtypes. That is, if a single mutation in one of the driver genes overcomes multiple barriers simultaneously, it will greatly accelerate the evolution of the malignant phenotype. However, in general, development of all lung cancers will be favored by an increased mutation rate which will allow them to generate the requisite mutations more quickly. This is consistent with increased risk of *KRAS*-mut, and NEK tumors associated with smoking while *EGFR*-mut cancers are frequently observed in non-smokers [58]. We note the mutational burden is increased by smoking [59] and a distinctive pattern of mutations is typically observed in smokers [20].

Our results have implications for the hypothesis that the “mutator phenotype” is essential for carcinogenesis. We find *EGFR*-mut cancers have a significantly lower mutational burden than the other subtypes and conserve 16 genes related to DNA repair compared to 1 and 3 in NEK and *KRAS*-mut subtypes, respectively. Extensive conservation of homeobox and RNA Polymerase II genes in the *EGFR*-mut cancers, suggests phenotypic plasticity is promoted through epigenetic modifications. In contrast, KRAS-mut and NEK tumors, which more commonly evolve in a mutagenic smoking environment, conserved fewer repair genes suggesting phenotypic heterogeneity is primarily driven by mutations.

While gene mutations are extensively investigated in cancers, the concept of conservation of normal genes under epigenetic control highlights the perhaps equally important role of conserved genes. Identifying genes under stabilizing selection can, thus provide novel insight into molecular pathways and cellular function critical for cancer cell survival and proliferation.

Finally, computer simulations (Appendix A) have predicted targeting conserved genes may reduce the tumor population as effectively as targeting driver genes [1]. However, if a conserved gene is also necessary for survival of normal cells, toxicity could be limiting. Our findings that most conserved genes are highly specific to each lung cancer type suggest that they could be targeted without significant impact on normal tissue. These evolutionary dynamics, termed co-adaptation, have potential clinical significance. Both *EGFR*-mut and *KRAS*-mut cancers are now being treated with targeted therapy. While often initially effective, most cancer populations succeed in evolving resistance leading to tumor progression. Often, resistance emerges from proliferation of a rare population with a pre-existent resistance mutation (e.g., T790m). However, even when that specific mutation is treated, *EGFR*-mut lung cancers access alternative genetic pathways to overcome therapy. Simulations of this process (Appendix A) show that conserved genes play a critical role in these evolutionary dynamics and predict that combination treatments targeting a driver gene and a driver-specific conserved gene may impose on the cancer a virtually unsolvable evolutionary conundrum that results in complete population loss (1). These findings, of course, are theoretical but may be used to guide future empirical studies.

## 5. Conclusions

The vast literature in cancer biology has identified extensive molecular changes in lung cancers. However, studies performed in vitro with cell lines are potentially limited by the ecological context of the experiments. That is, human cancer cells maintained in culture evolve to adapt to the environmental selection forces imposed by in vitro ecological conditions, which are vastly different from those in situ. As a result, cancer cells inevitably evolve to a “culture morph” so that its molecular properties are optimized for fitness in vitro. These eco-evolutionary dynamics generate uncertainty regarding the clinical relevance of pre-clinical experimental results.

Here, we apply Darwinian principles to the large publicly available data on molecular changes in clinical lung cancers. We use an inverse problem approach based on the hypothesis that observable molecular characteristics in multiple members of a tumor cohort represent common tumor cells’ evolutionary strategies for successful adaptation to intracellular and extracellular barriers encountered during carcinogenesis and tumor progression.

Initial computer simulations found the evolutionary arc of a tumor is sensitive to initial conditions, which include the molecular state of the initiating cell and the environmental properties of the local tissue in which somatic evolution takes place. By analyzing adenocarcinomas originating in the same tissue but with different initiating mutations, we separated these two dynamics. In general, our findings suggest each lung cancer population must overcome a relatively fixed number of intracellular and extracellular barriers to proliferation. Common mutational patterns among the 3 LUAD subtypes suggest they represent adaptations to these barriers. However, initiating mutations in genes with large interactome (e.g., *EGFR*, *TP53,* and *KRAS*) probably allow the cell to overcome multiple barriers simultaneously and are, therefore, favored. In the absence of a driver mutation, cancer cells need to accumulate far more mutations thus favoring mutagenic tissue environments.

On the other hand, conserved genes are unique to each cancer type and thus represent co-adaptations to the initiating mutations. In general, conserved genes represent the molecular wires or the downstream effectors of oncogenic signals and, thus, provide potentially valuable insights into critical molecular pathways and cellular functions. Computer simulation predict that disrupting tumor specific conserved genes can have a therapeutic benefit equal to or greater than targeted therapy for driver gene mutations.

## Figures and Tables

**Figure 1 cancers-15-00018-f001:**
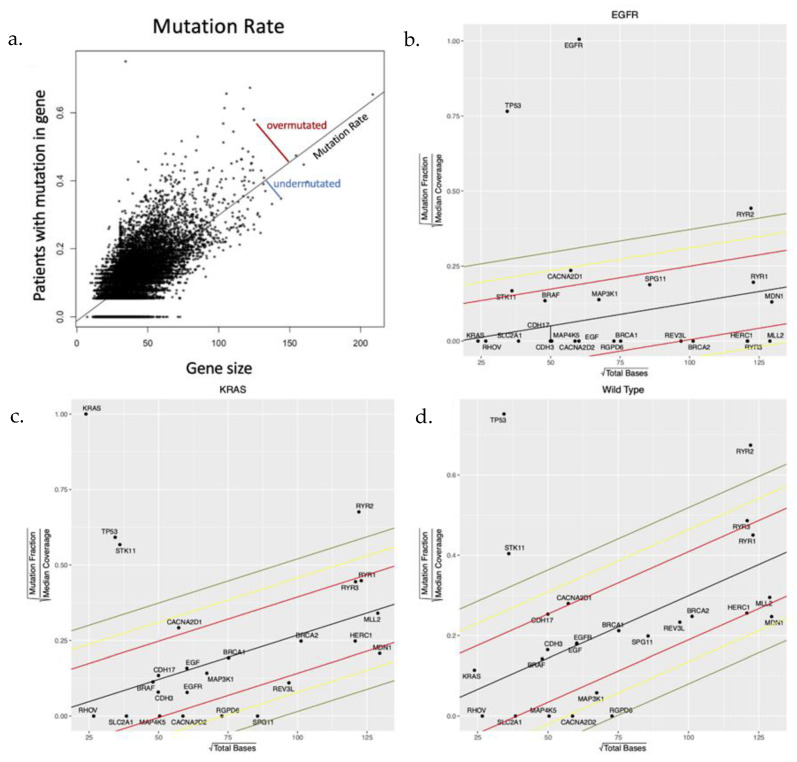
(**a**) A toy model demonstrating the methodology applied to the TCGA data set to identify genes that are observed to be mutated more or less frequently than expected by chance alone. (**b**–**d**) Simple regression of number of mutations observed in each gene compared to gene size (number of base pairs) produced a “neutral line” reflecting the number of genes mutated due to chance alone depending on the underlying mutation rate and gene size. The slope of the line in each cohort reflects the mutation rate which is smallest in the EGFR-mut cohort and largest in the NEK cohort. In each cohort, the genes of interest that are under evolutionary selection are shown.

**Figure 2 cancers-15-00018-f002:**
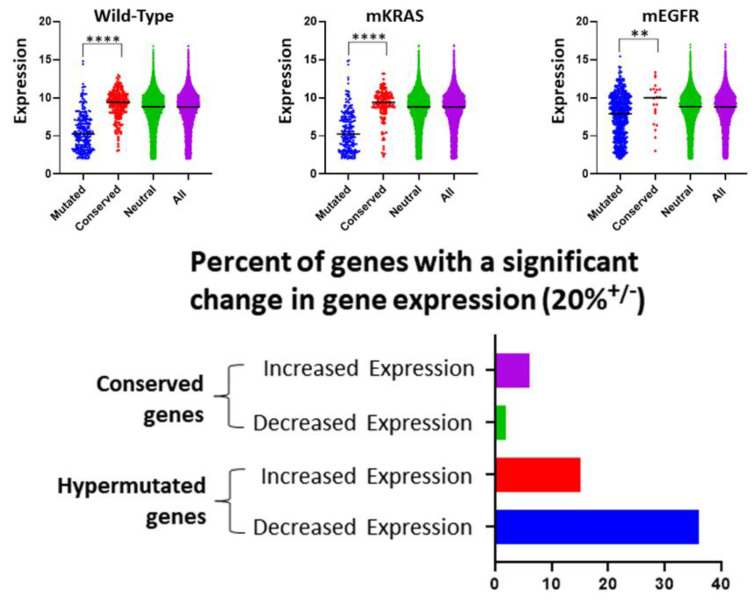
We address the role of gene expression in evolutionary selection. Upper Row. In all 3 cohorts, we find genes that are conserved (> 2 standard deviations below the neutral line [red]) have a median expression that is significantly higher (Welch’s *t*-test) than genes in which mutations are highly selected (> 2 standard deviation above the neutral line [blue]). For comparison, the distribution of expression in genes exhibiting a neutral selection pattern (i.e., < 2 standard deviation from neutral line [green]) and all genes (purple) are also shown. Note genes with expression < 2.0 were excluded from the mutational frequency analysis (see Section 2) and not included in the figures. Lower image. Over half of genes with evolutionarily selected mutations show > 20% increased or decreased expression. In the evolutionarily conserved genes, expression changes (particularly decreased expression) are uncommon. **** = *p* < 0.0001, ** = *p* < 0.01.

**Table 1 cancers-15-00018-t001:** Evolutionary selection on genes related to signaling pathways.

Evolutionary Selection for Mutations or Conservation in Intracellular Pathways					
*EGFR*-Mut		*KRAS*-Mut		NEK	
Conserved	Mutation	Conserved	Mutation	Conserved	Mutation
MAPK					
*HRASLS*	*MAP3K13*	*MAP3K6*	MAPK7	*MAP3K1*	*FAM83C*
*EGF*	*MAP7D2*	*MEGF11*	*FAM83B*	*MAP4K5*	
	*MAP2K4*	*VEGFC*	*INAVA*	*MEGF9*	
	*MAP3K11*				
	*MAP3K9*				
	*RASA2*				
	*RASAL2*				
	*RASL10B*				
MYC					
*HECTD4*					
NOTCH					
			*NOTCH4*		*NOTCH4*
Rho receptors					
*RHOV*	*RHOH*	*RHOV*		*RHOV*	
Rho/Rac guanine nucleotide exchange					
*ARHGEF19*	*ARHGEF40*	*ARHGEF5*	*ARHGEF6*	*ARHGEF1*	
	*ARHGAP6*	*ARHGAP36*	*ARHGAP6*	*ARGHEF39*	
		*KIAA0355* *(GARRE1)*		*ARHGEF39*	
Rab pathway					
*RAB26*		*RAB3GAP1*		*RAB3B*	
*RAB3GAP2*		*RAB26*			
		*TBC1D2B*		*TBC1D3H*	
				*TBC1D3C*	
		*DENND4C*			
		*DENND5A*			
			*FAM71B* *GARIN3*		*FAM71B* *GARIN3*
		*ST5*			
Rac pathway					
				*RAC3*	
RAN pathway					
				*RANBP9*	
Protein tyrosine phosphatase receptors					
*PTPRH*	*PTPRN*	*PTPRF*	*PTPRD*		*PTPRD*
			*PTPRT*		*PTPRT*
			*PTPRN*		*PTPRZ1*
ERBB					
	*ERBB4*		*ERBB4*		
Interleukin receptors					
*IL36RN*	*ILRL1*	*IL36RN*		*IL1R1*	
*IL23A*	*IL1RAPL2*	*IL17C*		*IL27RA*	
*IL37*				*IL23A*	
*IL31RA*					
*IL22RA2*					
*IL1RL2*					
*IL41L*					
*IL36G*					
Interferon					
				*IRF2BP1*	
				*IRF5*	
WNT/Catenin					
		*WNT3*			
			*CTNNA2*		*CTNNA2*
			*CTNND2*		*CTNND2*
			*FAM123B* *(AMER1)*		*FAM123C* *(AMER3)*
Lipid Receptors					
	*NLRP3*		*NLRP3*		*NLRP3*
	*NLRP14*		*NLRP14*		*NLRP14*
	*NLRP5*		*NLRP5*		*NLRP3*
			*LLRP1B*		*LRP1B*
			*NLRP10*		
					*LRP1B*
			*LRP1B*		
HIPPO					
		*WWC2*			
NOTCH					
			*NOTCH4*		*NOTCH4*
Fibroblast Growth Factor					
*FGF11*		*FGF11*		*FGF11*	
		*FGF19*			
Insulin like growth factors					
*IGF2BP3*		*IGFL2*		*IGFL2*	
				*IGFBP3*	
Toll Like Receptors					
			*TLR4*		*TLR4*
Ephrin Receptors					
*EPHX4*	*EPHA3*	*EPHA1*	*EPHA3*	*EPHA5*	*EPHA3*
*EPHB2*	*EPHA6*	*EPHB6*	*EPHA6*		*EPHA6*
			*EPHA5*		*EPHS5*
			*EPHB6*		
			*EPHA3*		
*EFNA4*		*EFNA4*			
*EFNA3*				*EFNA3*	
Ryanodine Receptors					
*RYR3*	*RYR2*		*RYR2*		*RYR2*
			*RYR3*		*RYR3*
			*RYR1*		*RYR1*
Glutamate Receptors					
	*GRIN2A*		*GRIN2B*		*GRIN2A*
					*GRIN2B*
			*GRID2*		*GRID2*
Semaphorin					
	*SEMA5A*				*SEMA5A*
	*SEMA5B*				*SEMA5B*
G-Coupled Protein					
*GPR115*	*GPR158*	*GPR113*	*GPR112*	*GPR108*	*GPR112*
*GPR87*			*GPR148*	*GPR19*	*GPR158*
*GPR110*				*GNL*	*GPR141*
					*GPR174*
					*GPR139*
					*GPR98*
					*GPR158*
Inositol					
*ITPKA*				*ITPKA*	
cAMP signaling					
	*ADCY2*		*ADCY2*		*ADCY2*
	*ADCY8*				*ADCY8*
					*ADCYAP*

**Table 2 cancers-15-00018-t002:** Evolutionary selection on genes related to phenotypic plasticity and epigenetic modifications.

Evolutionary Selection of Genes Associated with DNA Repair, Phenotypic Plasticity, and Epigenetic Modifications					
*EGFR*-Mut		*KRAS*-Mut		NEK	
Conserved	Mutation	Conserved	Mutation	Conserved	Mutation
Histones					
*HIST1H2AM*		*HIST1H2AM*			
*HIST1H2BD*		*HIST1H2BD*			
		*HIST1H2BG*			
Acetylation/Methylation					
*CDYL2*	*HDAC9*		*HDAC9*	*CDY1*	*HDAC9*
*ACY3*				*CDY1B*	
*BRD4*				*CDY2A*	
*B4GALNT44*				*CDY2B*	
*GCNT3*				*CLOCK*	
*ST8SIA2*				*HDAC5*	
				*HEXB*	
				*GCNT3*	
				*HDAC5*	
		*KIAA0182* *(GSE1)*			
Cytochrome P450 family					
*CYP2D6*					*CYP11B2*
*CYP27B1*					*CYP11B1*
*CYP27C1*					*CYP26B1*
*CYP241*					
					*ABCB1*
Homeobox/RNA Polymerase II					
*BARX2*		*BARX2*		*BARX2*	
*ETV4*		*ETV4*			
*SIX4*		*SIX1*			
*SIX2*					
*HOXC10*	*HOXA13*	*HOXB9*	*HOXA13*		
*HOXB9*	*HOXA1*	*HOXC10*			*HOXA1*
*HOXC13*	*HOXA3*	*HOXB3*			*HOXA3*
*HOXC9*		*HOXA10*			*HOXA5*
*HOXB13*					
*HOXD4*					
*HOXD3*					
*HOXB8*					
*HOXC11*					
*HOXC8*					
*HOXC6*					
*HOXB7*					
*FOXM1*	*FOXG1*		*FOXG1*		*FOXF2*
*FOXP3*					
*FOXB1*					
*ONECUT1*		*ONECUT1*			
*ONECUT2*		*ONECUT2*			
*OTX1*					
*PAX7*					*PAX4*
*PAX9*					
*PITX2*		*GREM1*			
*E2F2*					
*E2F8*					
		*ASXL1*	*ASXL3*		*ASXL3*
*POU3F2*	*POU3F4*				*POU3F4*
		*CIITA*			
		*NOMO2*		*NOMO3*	
			*SATB2*		*SATB2*
			*TSHZ3*		*TSHZ3*
			*ZEB1*		*ZEB1*
			*ZFHX4*		*ZFHX4*
Multicellularity					
*PLXNB3*	*SEMA5A*				*SEMA5A*
	*SEMA5B*			*SEMA4B*	*SEMA5B*
					*SEMA6D*
*FREM2*					*SEMA3D*
	*ABCB5*		*ABCB5*		*ABCB5*
					*ABCB1*
			*GLI2*		
*GINS1*		*GINS1*			
*GINS2*		*GINS2*			
E2F transcription factor 5					
*E2F8*					
*E2F2*					
Cyclin-dependent genes					
*CDKN3*	*CDKN2A*	*CDK13*	*CDKN2A*	*CDK5RAP1*	*CDKN2A*
		*CDKN3*		*CDKN3*	
		*CDK6*		*CDK5R2*	
		*CDKL2*		*CDK5R2*	
Telomere					
		*TERT*			
Zinc Finger proteins					
*ZFHX3*	*ZFPM2*	*ZNF729*	*ZNF479*	*NF729*	*ZNF479*
	*ZFHX4*	*ZNF827*	*ZNF536*	*ZNF324*	*ZNF536*
		*ZNF839*	*ZNF676*	*ZNF84*	*ZNF676*
		*ZNF687*	*ZNF804B*	*ZNF726*	*ZNF804B*
		*ZNF845*	*ZNF385D*	*ZNF749*	*ZNF385D*
		*ZC3H4*	*ZNF208*	*ZNF642*	*ZNF208*
		*PDZD8*	*ZNF257*	*ZNF30*	*ZNF257*
		*ZZEF1*	*ZNF716*	*ZNF57*	*ZNF716*
			*ZNF521*	*ZNF433*	*ZNF521*
			*ZNF831*	*ZNF655*	*ZNF831*
			*ZNF98*	*ZNF503*	*ZNF98*
			*ZNF804A*	*ZNF846*	*ZNF804A*
			*ZNF711*	*ZNF205*	*ZNF648*
			*ZNF679*	*ZNF26*	*ZBTB1*
			*ZNF835*	*ZFPM2*	*ZBBX*
			*NF479*	*TSHZ3*	*ZBTB1*
			*GLI3*	*ZFP106*	*ZFPM2*
			*ZEB1*	*ZBTB1*	*TSHZ3*
			*ZFPM2*	*ZDHHC5*	*ZEB1*
			*TSHZ3*	*TSHZ3*	*ZCCHC5*
			*ZIC4*		*ZIC4*
			*ZIC1*		*ZIC1*
			*ZSCAN4*		*ZIM2*
			*ZCCHC12*		*TSHZ2*
			*ZIC3*		*ZFHX4*
			*ZSCAN1*		
			*ZSCAN5B*		
			*ZFHX4*		
DNA repair					
*BRCA1*					
*BRCA2*					
*KIAA0101 (PCLAF)*		*KIAA0101* *(PCLAF)*			
*REV3L*		*REV3L*			
*XRCC2*					
*CHAF1B*					
*CLSPN*					
*EME1*		*EME1*			
*EXPO1*					
*FOXM1*					
*MCM10*					
*TONSL*					
*UBE2T*		*UBE2T*	*UBE2A*		
*UBE2C*					
*EXPO1*					
*HROB*					
				*AUNIP*	

**Table 3 cancers-15-00018-t003:** Evolutionary selection in genes associated with the extracellular matrix.

Extracellular Matrix					
*EGFR*-Mut		*KRAS*-Mut		NEK	
Conserved	Mutation	Conserved	Mutation	Conserved	Mutation
Matrix Metalloproteinase					
*MMP17*			*MMP16*		*MMP16*
*MMP9*			*MMP2*		
ADAMTS					
	*ADAMTS5*	*ADAMTS8*	*ADAMTS16*		*ADAMTS16*
	*ADAMTS14*		*ADAMTS2*		*ADAMTS2*
			*ADAMTS20*		*ADAMTS20*
			*ADAMTS12*		*ADAMTS12*
			*ADAMTS18*		
Membrane anchored proteases					
*TMPRSS4*	*TMPRSS12*	*TMPRSS4*	*TMPRSS11E*		
*TMPRSS11E*	*TMPRSS15*		*TMPRSS15*		
Bone Marrow Morphogenetic Proteins					
*BMP8A*	*BMP1*				
	*FAM5C* *(BRINP3)*		*FAM5C* *(BRINP3)*		*FAM5C* *(BRINP3)*
					*FAM5B* *(BRINP2)*
Protease inhibitors					
*CST1*		*CST1*			
*CST2*		*CST2*			
*CST4*					
*SPINK1*		*SPINK1*		*SPINK1*	
*SPINK13*		*SPINK13*			
*SPINK2*					
	*SERPINA7*		*SERPINB4*		*SERPINB3*
	*SERPINI1*		*SERPINA5*		
Collagen					
*COL1A1*	COL6A2		*COL6A2*	*COL4A3BP*	*COL22A1*
*COL9A2*	COL7A1		*COL22A1*		*COL19A1*
*COL10A1*	COL23A1		*COL19A1*		*COL25A1*
*COL24A1*			*COL25A1*		*COL3A1*
			*COL3A1*		*COL11A1*
			*COL11A1*		*COL5A2*
			*COL21A1*		*COL14A1*
			*COL1A2*		*COL6A3*
			*COL6A3*		*COL3A1*
					*COL6A6*
					*COL12A1*
Tenascin					
	ODZ1		*ODZ1*		*ODZ1 (TENM3)*
	ODZ2		*ODZ3*		*ODZ3*
	TNR		*TNR*		*TNR*
			*TNN*		*TNN*
Other ECM components					
	*FLG*		*FLG*		*FLG*
			*FLG2*		*FLG2*
Mucin					
*MUC21*	*MUC17*		*MUC16*		*MUC17*
*MUC13*	*MUC16*				*MUC7*
					*MUC5B*
Laminin Beta Subunit					
	*LAMB4*				
Protocadherin					
	*PCDHB2*		*PCDHB2*	*PCDHGB5*	*PCDHB2*
	*PCDH15*		*PCDH15*		*PCDH15*
	*PCDHGA2*		*PCDHGA2*		*PCDHGA2*
	*PCDHB7*		*PCDHB7*		*PCDHB7*
	*PCDHGB2*		*PCDHGB2*		*PCDHB11*
	*PCDHA4*		*PCDHA4*		*PCDHA2*
	*PCDHGB3*		*PCDHGB3*		*PCDHB4*
	*PCDHB8*		*PCDHB11*		*PCDH11X*
	*PCDHA1*		*PCDHA2*		*PCDH10*
	*PCDHGA3*		*PCDHB4*		*PCDHA3*
	*PCDHGA7*		*PCDH11X*		*PCDH178*
	*PCDHGA1*		*PCDH10*		*PCDHB12*
	*PCDHGB4*		*PCDHA3*		*PCDH18*
	*PCDHGC5*		*PCDH178*		*PCDHB14*
	*PCDHGC4*		*PCDHB12*		*PCDH11Y*
	*PCDHGA10*		*PCDHA6*		*PCDH8*
	*PCDHGA5*				*PCDHB3*
	*PCDHGA9*				*PCDH17*
	*PCDHGA12*				
	*PCDHGA4*				
	*PCDHB6*				
	*PCDHGB*				
	*PCDH19*				
	*PCDHGB5*				
	*PCDHGA8*				
	*PCDHGA11*				
	*PCDHGB6*				
	*PCDHGC3*				
	*PCDHGA6*				
	*PCDHGB1*				
Cadherins					
*CDH3*	*CDH10*		*CDH10*		*CDH10*
*CDH17*	*CDH6*		*CDH18*		*CDH6*
	*CDH13*		*CDH9*		*CDH18*
	*CDHR1*		*CDH22*		*CDH9*
	*CDHR2*		*CDH7*		*CDH22*
	*CDH18*		*CDH8*		*CDH7*
	*CDH23*		*CDH2*		*CDH12*
			*CDH11*		
Atypical cadherins					
*FAT4*			*FAT3*		*FAT3*
			*FAT4*		*FAT4*
					*FAT1*
Myosin					
*MYO7A*	*MYO7B*			*MYO5C*	*MYO7B*
	*MYOD1*				*MYO18B*
	*MYO18A*				
	*MYO1G*				
Perlecan proteins					
*HSPG2*					
Serine/threonine kinase					
*STK32A*			*STK11*		*STK11*
Intergrins					
		*INHA*			
		*IBSP*			

**Table 4 cancers-15-00018-t004:** Evolutionary selection in genes associated with cell membrane.

Evolutionary Selection for Mutations or Conservation of Genes Associated with the Cell Membrane					
*EGFR*-Mut		*KRAS*-Mut		NEK	
Conserved	Mutation	Conserved	Mutation	Conserved	Mutation
Claudin					
*CLDN3*	*CLDN15*	*CLDN1*			
*CLDN2*					
*CLDN6*					
*CLDN9*					
Gap junctions					
*GJB2*		*GJB2*		*GJB2*	
*GJB6*		*GJB6*			
		*GJB3*			
Spectrin family					
*SPTBN5*					
*SPTBN2*					
Cilia					
*DNAH10*			*DNAH11*		*DNAH11*
			*DNAH9*		*DNAH9*
					*DNAH8*
					*DNAH3*
					*DNAH7*
					*DNAH5*
				*BBOF1*	
		*KIAA0753*			
		*IFT122*			
*SPAG5*	*SPAG11B*			*SPAG4*	
*SPAG9*	*SPAG4*				
*NEK2*				*NEK2*	
	*C1orf174* *ERICH3*		*C1orf174* *ERICH3*		*C1orf174* *ERICH3*
*C16orf59* *(TEDC2)*		*C16orf59* *(TEDC2)*		*C16orf59* *(TEDC2)*	
Ca^++^ Voltage gated channels					
*CBARP*	*CACNA2D1*	*CACNA2D2*	*CACNA2D1*	*CACNA2D2*	*CACNA2D1*
	*CACNA1E*		*CACNA1E*		
	*CACNA1C*		*CACNA1C*		*CACNA1C*
			*CACNA2D3*		
K^+^ Voltage gated channels					
*KCNQ5*	*KCNK2*	*KCNN4*	*KCNK2*	*KCNS1*	*KCNK2*
	*KCNA5*		*KCNA5*	*KCNC3*	*KCNA5*
	*KCNH1*		*KCNH1*		*KCNK10*
	*KCNH8*		*KCNJ18*		*KCNT2*
	*KCNH5*		*KCNA4*		*KCNK13*
			*KCNB2*		
			*KCNJ12*		
			*KCNJ3*		
			*KCNU1*		
			*KCNC2*		
			*KCNJ2*		
			*KCNS2*		
			*KCNQ2*		
K^+^ channel tetramerization domain-containing family					
		*KCTD19*			*KCTD8*
Hyperpolarization-activated cyclic nucleotide-gated potassium channel					
			*HCN1*		*HCN1*
Na^+^ channels					
			*SCN3A*		*SCN3A*
					*SCN2A*
Sodium leak Channel, non-selective					
			*NALCN*		*NALCN*
Acid sensing channel					
					ASIC2
Glutamate-gated ion channel					
	*GRIN2A*				*GRIN2A*
			*GRIN2B*		*GRIN2B*
					*GRIN3A*
Cancer-testis genes					
*XAGE1B*		*XAGE1B*		*XAGE1B*	
*XAGE1D*		*XAGE1D*		*XAGE1D*	
	*MAGEH1*		*MAGEC1*	*MAGED4*	*MAGEC1*
	*MAGED2*		*MAGEC3*	*MAGED4B*	*MAGEC3*
	*MAGED1*		*MAGEC2*	*MAGEA2*	*MAGEC2*
			*MAGEB4*		*MAGEB4*
			*MAGEA10-MAGEA5*		*MAGEB18*
			*MAGEB16*		
			*MAGEA5*		
			*MAGEA6*		
			*MAGEA4*		
*CXorf61* *CT83*		*CXorf61* *CT83*		*CXorf61* *CT83*	
Spermatogenesis					
			*FAM75D1* *(SPATA31)*		*FAM75D1* *(SPATA31)*
		*FAM75A4* *(SPATA31A)*		*FAM75A4* *(SPATA31A7)*	*FAM75A6* *(SPATA31A6)*
		*FAM75C2* *(SPATA31C2)*		*FAM75C2* *(SPATA31C2)*	
					*FAM75A5* *(SPATA31A5)*
	*C15orf2* *(NPAP1)*		*C15orf2* *(NPAP1)*		*C15orf2* *(NPAP1)*
	*CXorf59* *(CFAP47)*		*CXorf59* *(CFAP47)*		*CXorf59* *(CFAP47)*
Fucosyltransferases					
*FUT2*				*FUT4*	
*FUT9*				*FUT2*	
*FUT3*					
*FUT6*					
Solute Carriers					
*SLC1A7*	*SLC5A11*	*SLC2A1*	*SLC6A5*	*SLC1A4*	*SLC8A1*
*SLC2A1*	*SLC6A13*	*SLC7A5*	*SLC6A15*	*SLC2A1*	*SLC10A2*
*SLC6A11*	*SLC6A17*	*SLC7A10*	*SLC8A1*	*SLC7A8*	*SLC12A1*
*SLC6A3*	*SLC10A1*	*SLC24A1*	*SLC8A3*	*SLC7A11*	*SLC17A3*
*SLC7A10*	*SLC34A2*	*SLC26A8*	*SLC9A4*	*SLC19A1*	*SLC17A6*
*SLC15A1*	*SLCO1B1*		*SLC17A6* *SLC17A6*	*SLC22A5*	*SLC35F1*
*SLC16A9*	*SLCO4A1*		*SLC27A6*	*SLC34A3*	*SLC39A12*
*SLC29A4*			*SLC30A10*	*SLC44A2*	*SLC44A5*
*SLC52A1*			*SLC39A12*	*SLC44A4*	
*SLCO5A1*					
Transmembrane proteins					
*TMEM184A*	*TMEM200A*	*TMEM184A*	*TMEM200A*	*TMEM180*	*TMEM196*
*TMEM63C*				*TMEM201*	*TMEM132*
*TMEM156*					
*TMEM59L*					
ATP-Binding Cassettes					
*ABCA4*	*ABCB5*		*ABCB5*	*ABCF1*	*ABCB5*
*ABCC3*			*ABCA13*		*ABCB1*
		*ATP2A1*			*ABCA13*
		*ATP8B3*			
		*ATP2A2*			

## Data Availability

All data is included in the figure and tables or Appendix A.

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
