# Peer review of "Evolutionary Analysis of TCGA Data Using Over- and Under- Mutated Genes Identify Key Molecular Pathways and Cellular Functions in Lung Cancer Subtypes"

_cancers, 2022, doi:10.3390/cancers15010018_

Round 1

Reviewer 1 Report

Freischel et al. cancers-2014646 " Evolutionary analysis of TCGA data using over- and under- mutated genes identify key molecular pathways and cellular functions in lung cancer subtypes" is a valuable review paper because the authors used RNA-seq and TCGA-based analysis to identify conserved or mutated genes in three LUAD mutants (EGFR-mut, KRAS-mut, and NEK). Furthermore, the authors found differences in gene expression in these cells. However, some points are difficult for the reviewer to understand. The reviewer hopes that providing more information (described below) will improve the quality of this study. 

1.     This study identified many conserved and mutated genes in the three LUAD variants (EGFR-mut, KRAS-mut, and NEK) by RNA-seq and TCGA analysis, and further showed that there were differences. However, the reviewer feels that there is no specific mention in the paper of how to treat LUADs with mutated genes by exploiting the differences in gene expression in these cells. Therefore, the reviewer believes that this description is an essential aspect of the paper published in CANCERS.

2.     The reviewer believes that Supplemental Figure 5 is an important result of this study. Therefore, the reviewer believes that the results in Supplemental Figure 5 should be moved to the Results section to discuss the results in detail, and future issues based on the results should be discussed in the Discussion section.

Author Response

Response to Reviewers

Reviewer 1

Open Review

English language and style

( ) English very difficult to understand/incomprehensible
( ) Extensive editing of English language and style required
( ) Moderate English changes required
( ) English language and style are fine/minor spell check required
(x) I don't feel qualified to judge about the English language and style

Yes

Can be improved

Must be improved

Not applicable

Does the introduction provide sufficient background and include all relevant references?

(x)

( )

( )

( )

Are all the cited references relevant to the research?

(x)

( )

( )

( )

Is the research design appropriate?

( )

(x)

( )

( )

Are the methods adequately described?

(x)

( )

( )

( )

Are the results clearly presented?

( )

(x)

( )

( )

Are the conclusions supported by the results?

( )

(x)

( )

( )

Comments and Suggestions for Authors

Freischel et al. cancers-2014646 " Evolutionary analysis of TCGA data using over- and under- mutated genes identify key molecular pathways and cellular functions in lung cancer subtypes" is a valuable review paper because the authors used RNA-seq and TCGA-based analysis to identify conserved or mutated genes in three LUAD mutants (EGFR-mut, KRAS-mut, and NEK). Furthermore, the authors found differences in gene expression in these cells. However, some points are difficult for the reviewer to understand. The reviewer hopes that providing more information (described below) will improve the quality of this study. 

  1. This study identified many conserved and mutated genes in the three LUAD variants (EGFR-mut, KRAS-mut, and NEK) by RNA-seq and TCGA analysis, and further showed that there were differences. However, the reviewer feels that there is no specific mention in the paper of how to treat LUADs with mutated genes by exploiting the differences in gene expression in these cells. Therefore, the reviewer believes that this description is an essential aspect of the paper published in CANCERS.

Thank you. This is a good point and we have added more details about this aspect of the work in the Discussion section.

  1. The reviewer believes that Supplemental Figure 5 is an important result of this study. Therefore, the reviewer believes that the results in Supplemental Figure 5 should be moved to the Results section to discuss the results in detail, and future issues based on the results should be discussed in the Discussion section.

 Thank you. We have now added a paragraph at the end of the Discussion to specifically address targeted therapy in KRAS-mut and EGFR-mut lung cancers.  In particular, we note that computer simulations have suggested the change in “molecular wiring” of cancer cells to adapt to loss of oncogenic signals from the driver gene critically depends on driver-specific conserved genes. As a result, simulations suggest the combination of targeted therapy for the driver gene with disruption of a driver-associated conserved gene could be a highly effective treatment. Of course, we note these are theoretical studies that are meant to guide future empirical studies.

Reviewer 2 Report

This is an interesting manuscript presenting an evolutionary analysis of mutated and conserved genes in  EGFR-mutated, KRAS-mutated, and non-EGFR-/KRAS-mutated lung adenocarcinomas (LUADs) using TCGA data. The authors show a significant overlap of frequently mutated genes among the three subtypes and interpret this as directional selection to counteract the growth restrictions present in normal lung tissue. In contrast, highly expressed genes are shown to be usually conserved, most likely because they are essential for the fitness and growth of cancer cells. Finally, by analyzing the variation in conserved and mutated genes, the authors describe key interactome pathways specific for each of the three subtypes. The results have relevant implications for understanding the evolutionary molecular biology and novel vulnerabilities of LUADs that may be exploited for future targeted therapies.

Yet, a minor revision with some adjustments and elucidations seems required to make the MS fully publishable.

SPECIFIC POINTS

Introduction

Line 36-37, “We use differences in patterns of Darwinian selection for critical genes in EGFR-mut, KRAS-mut, and NEK (non-EGFR/KRAS group) lung cancers …”: The abbreviations should be defined here (and in the Abstract too), not on line 91-92.

Line 84-85, “In contrast, we find only a small number of genes with an increased mutation rate in all the LUAD subtypes …”: the abbreviation LUAD should be defined here, not on line 89.

Methods

Line 101, “We divided the TCGA lung adenocarcinoma cohort”: to be consistent, the abbreviation LUAD should be used here.

Line 121-122, “Somatic mutations from TCGA LUAD samples (n=534 patients) were divided into EGFR-mut (n=58), KRAS-mut (n=163), and NEK (n=313)”: this is a repetition of the senntence written on line 105-106 and should be eliminated.

Line 124-126, “Gene-level depth of coverage was determined by calculating the number of bases covered by sequencing for each of the RefSeq coding genes (with 25 base pair flanking regions)”: all this has already been stated on line 108-110.

Caption to Supplemental Table 1:  For the sake of clarity, it should be better explained why "Higher standard deviations represent the most likely gene mutation under positive selection".

Results

Important demographic data (that should be available from the TCGA db), such as age, gender, smoking status, and tumor stage of the examined cases should be indicated for each of the three cohorts (for ex. by a table), as these variables may impact mutation rate and gene transcription/translation rate. As also mentioned by the authors in the Discussion, EGFR-mutations occur in LUAD patients who are non-smokers and younger than the average lung cancer patient. Thus, further correlation of the authors results with the above-mentioned demographic variables should presented. That is, do they have an impact on the results of the study?

Line 190-191, “highly expressed genes are also highly conserved” vs. 194-195 “highly conserved genes rarely show a significant change in expression (compared to normal lung tissue)”. At a first glance, the two sentences seem to contradict each other. Should they be rephrased to avoid misunderstandings?

Figure 1: For clarity, the letters a-d mentioned in the legend should also be indicated on the figure. Moreover, names of the coordinates on X and Y axis and corresponding units should be indicated in each figure.

Figure 2:  for clarity, bar colors should be indicated in the text of the figure legend too. Right now, it is difficult to follow the text on the figures.

Line 300-301, “indicating mutant P53 retains some functions that promote the proliferation and survival of the cancer cells”: this assumption could be modulated a little bit, as the authors have not analyzed whether p53 has retained some of its functions in the cancer cells of the analyzed LUAD patients.

Line 320-322, “Upregulation of EGF may support an autocrine loop that activates complementary signaling circuits to enhance proliferation and/or survival although this has not been previously investigated”:  the statement sounds speculative, do the authors have any evidence for that from their or others' data/publications?

The data on mutated and conserved genes and corresponding interactome analysis suggested by table 1-4 are interesting and may deserve further considerations regarding the evolutionary gene selection in the three cohorts. For instance, it would be appropriate to comment on the result of CDKN2A being mutated in all 3 groups. This is somehow consistent with previously published TCGA data showing that this gene is silenced in up to 40% of LUADs (DOI: 10.1038/nature13385). What would be the biological interpretation? That CDKN2A functions as restriction that needs to be overcome by cancer cells in all three subtypes or else?

Similarly, the status for ABC genes shown in table 4 could have been mentioned and further elaborated in the text, as this gene family codes for proteins functioning as efflux pumps for several substrates such as xenobiotics, phospholipids and drugs and may therefore mediate the development of drug resistance in tumor cells.

Discussion

Line 442-444, “However, we acknowledge, however, that other co-variates (56) could introduce produce intra-genomic variations in mutation rate that may introduce errors in our investigation”: One “however” and “introduce” should be deleted.

Line 464-465, “In other words, a malignant lung cancer requires …”: the adjective “malignant” seems unnecessary, as cancer is malignant by definition.

Line 476-478, “We hypothesize a requirement for more mutations to address a fixed number of environmental selection forces accounts for observations that EGFR-mut LUADs occur in younger patients than the other subtypes”: The sentence should be reformulated more clearly.

Should the Supplementary Fig. 5 be understandable in a clinical context for the broad readership of Cancers, then it should be simplified (both figure and figure legend should be simpler).

The above-mentioned suggestions should be easily addressed by the Authors. Thank you for the opportunity to review this work. 

Author Response

Review Report Form

Open Review

English language and style

( ) English very difficult to understand/incomprehensible
( ) Extensive editing of English language and style required
( ) Moderate English changes required
(x) English language and style are fine/minor spell check required
( ) I don't feel qualified to judge about the English language and style

Yes

Can be improved

Must be improved

Not applicable

Does the introduction provide sufficient background and include all relevant references?

(x)

( )

( )

( )

Are all the cited references relevant to the research?

(x)

( )

( )

( )

Is the research design appropriate?

(x)

( )

( )

( )

Are the methods adequately described?

( )

(x)

( )

( )

Are the results clearly presented?

( )

(x)

( )

( )

Are the conclusions supported by the results?

(x)

( )

( )

( )

Comments and Suggestions for Authors

This is an interesting manuscript presenting an evolutionary analysis of mutated and conserved genes in  EGFR-mutated, KRAS-mutated, and non-EGFR-/KRAS-mutated lung adenocarcinomas (LUADs) using TCGA data. The authors show a significant overlap of frequently mutated genes among the three subtypes and interpret this as directional selection to counteract the growth restrictions present in normal lung tissue. In contrast, highly expressed genes are shown to be usually conserved, most likely because they are essential for the fitness and growth of cancer cells. Finally, by analyzing the variation in conserved and mutated genes, the authors describe key interactome pathways specific for each of the three subtypes. The results have relevant implications for understanding the evolutionary molecular biology and novel vulnerabilities of LUADs that may be exploited for future targeted therapies.

Yet, a minor revision with some adjustments and elucidations seems required to make the MS fully publishable.

We thank the reviewer for his/her comments and appreciate the careful reading of the manuscript. As demonstrated below we have modified the manuscript to address the concerns

SPECIFIC POINTS

Introduction

Line 36-37, “We use differences in patterns of Darwinian selection for critical genes in EGFR-mut, KRAS-mut, and NEK (non-EGFR/KRAS group) lung cancers …”: The abbreviations should be defined here (and in the Abstract too), not on line 91-92.

Abbreviations defined at beginning of introduction, removed from lines 91-92

Line 84-85, “In contrast, we find only a small number of genes with an increased mutation rate in all the LUAD subtypes …”: the abbreviation LUAD should be defined here, not on line 89.

Abbreviation definition added here

Methods

Line 101, “We divided the TCGA lung adenocarcinoma cohort”: to be consistent, the abbreviation LUAD should be used here.

Replaced wording with abbreviation

Line 121-122, “Somatic mutations from TCGA LUAD samples (n=534 patients) were divided into EGFR-mut (n=58), KRAS-mut (n=163), and NEK (n=313)”: this is a repetition of the senntence written on line 105-106 and should be eliminated.

Line 124-126, “Gene-level depth of coverage was determined by calculating the number of bases covered by sequencing for each of the RefSeq coding genes (with 25 base pair flanking regions)”: all this has already been stated on line 108-110.

Thank you. The redundancy has been removed through re-arrangement of section.

Caption to Supplemental Table 1:  For the sake of clarity, it should be better explained why "Higher standard deviations represent the most likely gene mutation under positive selection".

Results

Important demographic data (that should be available from the TCGA db), such as age, gender, smoking status, and tumor stage of the examined cases should be indicated for each of the three cohorts (for ex. by a table), as these variables may impact mutation rate and gene transcription/translation rate. As also mentioned by the authors in the Discussion, EGFR-mutations occur in LUAD patients who are non-smokers and younger than the average lung cancer patient. Thus, further correlation of the authors results with the above-mentioned demographic variables should presented. That is, do they have an impact on the results of the study?

This is a great point. Unfortunately, the demographic data available through the TCGA is missing or incomplete in about 20% of the members in each cohort. We have added a paragraph to the results section with details. Briefly, patient age and tumor stage at presentation did not significantly among the cohorts. Consistent with multiple prior studies, patients with EGFR-mut LUAD were far more likely to be female and lifelong non-smokers compared to the other cohorts.

Line 190-191, “highly expressed genes are also highly conserved” vs. 194-195 “highly conserved genes rarely show a significant change in expression (compared to normal lung tissue)”. At a first glance, the two sentences seem to contradict each other. Should they be rephrased to avoid misunderstandings?

Thank you. Although the phrasing is bit awkward both statements are correct. We have amended this for clarity.

Figure 1: For clarity, the letters a-d mentioned in the legend should also be indicated on the figure. Moreover, names of the coordinates on X and Y axis and corresponding units should be indicated in each figure.

Done, thank you.

Figure 2:  for clarity, bar colors should be indicated in the text of the figure legend too. Right now, it is difficult to follow the text on the figures.

Done. Thank you. The senior author is color-blind but has been assured the current color designations are correctly state in the tex.

Line 300-301, “indicating mutant P53 retains some functions that promote the proliferation and survival of the cancer cells”: this assumption could be modulated a little bit, as the authors have not analyzed whether p53 has retained some of its functions in the cancer cells of the analyzed LUAD patients.

Thank you. We have amended this to say the data  “suggest” that P53 “may” retain some functions” to emphasize it is an hypothesis and have included a reference that is supportive.

Line 320-322, “Upregulation of EGF may support an autocrine loop that activates complementary signaling circuits to enhance proliferation and/or survival although this has not been previously investigated”:  the statement sounds speculative, do the authors have any evidence for that from their or others' data/publications?

Yes, we extensively search prior literature and could find no studies suggesting EGF has an ongoing role in EGFR-mut lung cancers, nor could we find evidence to the contrary. We have amended the text for clarity and suggested this may be a topic for future experimental studies.

The data on mutated and conserved genes and corresponding interactome analysis suggested by table 1-4 are interesting and may deserve further considerations regarding the evolutionary gene selection in the three cohorts. For instance, it would be appropriate to comment on the result of CDKN2A being mutated in all 3 groups. This is somehow consistent with previously published TCGA data showing that this gene is silenced in up to 40% of LUADs (DOI: 10.1038/nature13385). What would be the biological interpretation? That CDKN2A functions as restriction that needs to be overcome by cancer cells in all three subtypes or else?

Similarly, the status for ABC genes shown in table 4 could have been mentioned and further elaborated in the text, as this gene family codes for proteins functioning as efflux pumps for several substrates such as xenobiotics, phospholipids and drugs and may therefore mediate the development of drug resistance in tumor cells.

We thank the reviewer and are truly delighted (!) that he/she raised these issues. This is exactly the purpose of this work – to allow evolution to highlight genes, molecular pathways, and cellular functions that warrant more detailed investigation. We have tried to clarify in the text that this manuscript can, at best, scratch the surface of the results.

Discussion

Line 442-444, “However, we acknowledge, however, that other co-variates (56) could introduce produce intra-genomic variations in mutation rate that may introduce errors in our investigation”: One “however” and “introduce” should be deleted.

Thank you. We have amended the relevant text

Line 464-465, “In other words, a malignant lung cancer requires …”: the adjective “malignant” seems unnecessary, as cancer is malignant by definition.

Thank you, the word “malignant” has been removed

Line 476-478, “We hypothesize a requirement for more mutations to address a fixed number of environmental selection forces accounts for observations that EGFR-mut LUADs occur in younger patients than the other subtypes”: The sentence should be reformulated more clearly.

Thank you. We have amended that sentence and added some additional material in the conclusion

Should the Supplementary Fig. 5 be understandable in a clinical context for the broad readership of Cancers, then it should be simplified (both figure and figure legend should be simpler).

This is a good point and we have tried to simplify the figure through a clearer explanation in the text.  We also refer the readers to the initial publication in Nature Communication which discusses this in far more detail.

The above-mentioned suggestions should be easily addressed by the Authors. Thank you for the opportunity to review this work. 

Round 2

Reviewer 1 Report

The second revised paper seems to improve and should be worth publishing in this journal.